# Zeolite-Containing Co Catalysts for Fischer–Tropsch Synthesis with Tailor-Made Molecular-Weight Distribution of Hydrocarbons

**Lilia Sineva [1], Vladimir Mordkovich [1,*] , Ekaterina Asalieva [1] and Valeria Smirnova [1,2]**

[1]  Technological Institute for Superhard and Novel Carbon Materials, Troitsk, 108840 Moscow, Russia; sinevalv@tisnum.ru (L.S.); smirnova.ve@phystech.edu (V.S.)
[2]  Department of Physics and Chemistry of Nanostructures, The Moscow Institute of Physics and Technology, Dolgoprudny, 141701 Moscow, Russia
*  Correspondence: mordkovich@tisnum.ru; Tel.: +7-9166490738

**Abstract:** The review is dedicated to the topical field of research aimed at creating catalysts combining several types of active sites. At the same time, the composition of Fischer–Tropsch synthesis (FTS) products can be controlled by changing the strength and concentration of the active sites and inter-site distances. A comparative analysis of the literature data allows to formulate the main principles of catalytic particles formation active in FTS and acid-catalyzed transformations of hydrocarbons: (1) the presence of weak Bronsted acid sites to control the cracking depth, (2) an availability of Bronsted acid sites for re-adsorption hydrocarbons and (3) weak Co-zeolite interaction to reduce methane formation.

**Keywords:** Fischer-Tropsch synthesis; bifunctional catalysts; zeolite; molecular weight distribution; secondary transformations; active sites

## 1. Introduction

Hydrocarbons obtained through FTS from a mixture of CO and $H_2$, also known as synthesis gas, are one of the promising sources of hydrocarbon feedstocks for further use in chemical and petrochemical industries [1–4]. The composition of hydrocarbon mixtures depends on both the catalyst properties and the process parameters (Figure 1). The products of FTS usually follow the molecular weight distribution (MWD) [1–6], which is very wide and nonselective for any products being determined by the competition between the processes of growth and hydrocarbon chain termination. For example, high molecular weight n-paraffins are selectively formed in the presence of the Co-based catalysts on alumina, silica or titania supports. Such a mixture of hydrocarbons should be further processed at high temperature and pressure in excess of hydrogen for future use.

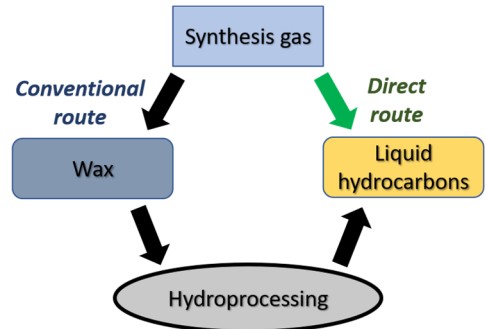

**Figure 1.** Potential routes of syngas-to-liquid hydrocarbon process based on Fischer–Tropsch synthesis.

An alternative process is FTS aimed at the direct production of liquid hydrocarbons (synthetic oil) from CO and $H_2$ [6–9]. The idea of combining metal sites active in FTS with acid sites of zeolite active in the cracking and isomerization of hydrocarbons was proposed in the middle of 1970s and continues to be of keen interest to the scientific community [6–16]. Such catalysts are routinely referred to as bifunctional or hybrid. The composition of the hydrocarbons obtained in their presence depends on the relative rates of reactions on acid and metal sites. The latter depend on concentrations of metal and zeolite sites, their location relative to each other and properties of the support pore system. Therefore, one has to consider the role of mass transfer while developing new bifunctional FTS catalysts [17]. In order to provide intensive mass transfer on the surface of the pellet, it is necessary to reduce diffusion limitations or average molecular weight of the formed hydrocarbons. In the first case, an extended system of macro- and meso-pores is necessary; in the second case, a source of atomic hydrogen or an additional function of the catalyst, —for example, provided by sites with Bronsted acidity—is necessary.

In order to obtain hydrocarbons with a narrower MWD, it is necessary to study both the mutual influence of the components of bifunctional catalysts and the location of metal/acid sites. The composition of FTS products can be controlled by changing the number or/and strength of acid sites that are determined by Si/Al ratio [13].

Therefore, this review is dedicated to the topical field of research aimed at creating catalysts combining several types of active sites. The composition of FTS products can be controlled by changing the strength and concentration of the active sites and inter-site distances.

## 2. The Role of Cobalt in the Formation of Fischer–Tropsch Synthesis Products

Thermodynamic calculations show that hydrocarbons of any molecular weight, type and structure except acetylene can be formed from CO and $H_2$ in the presence of Co-containing catalysts [1–4,18]. However, the FTS contains a lot of consecutive and parallel reactions and thermodynamic calculations based on the assumptions about concurrent equilibrium, so it only allows to evaluate the probability of products formation approximately. In addition, the rates of each reaction depend on the process parameters: for example, with an increase in temperature, the probability of forming unsaturated hydrocarbons and aldehydes increases, and the probability of forming saturated hydrocarbons decreases. With an increase in pressure, the content of heavy hydrocarbons increases. At low-temperature FTS in the presence of Co and Fe catalysts, the fraction of branched products up to $C_{14}$ does not vary significantly with molecular mass and hydrogen saturation [19]. An increase in the hydrogen content in the synthesis gas favors the formation of saturated linear hydrocarbons. In the case of CO content, an increase favors the formation of olefins and aldehydes, while the degree of carbonization of the catalyst increases. As a result, the actual composition of FTS products differs significantly from thermodynamic calculations.

The main FTS reactions correspond to the polymerization mechanism where the single fragment is $CH_x$ monomer formed by interaction of CO and $H_2$ [1–4,18,20]:

$$nCO + (2n + 1)H_2 \rightarrow C_nH_{2n+2} + nH_2O \tag{1}$$

$$nCO + 2nH_2 \rightarrow C_nH_{2n} + nH_2O \tag{2}$$

Side reactions in the case of low-temperature FTS in the presence of Co catalysts are as follows: formation of methane by direct CO hydrogenation; and water gas shift reaction. In the case of high-temperature FTS, the Boudouard reaction is a predominant side reaction. In addition, CO and $H_2$ can interact to form oxygen-containing compounds such as alcohols [1,2,4,20], although the probability of their formation is higher in the presence of Fe than Co. It is necessary to take into account the activity of metal sites in the secondary transformation of FTS-generated hydrocarbons such as hydrogenation, re-adsorption of α-olefins followed by their subsequent inclusion in the chain growth, hydrogenolysis and isomerization [9,21–24].

Secondary transformations of hydrocarbons depend on the carbon chain length and can have an effect on the composition of the resulting products. Kuipers E.W. et al. [21] used flat model catalysts—a poly-crystalline cobalt foil and Co particles deposited on $SiO_2$—since it seemed easier for interpretation of experimental results than in the case of porous system modeling. Hydrogenation of olefins was the main secondary reaction on cobalt foil and depended on the chain length. As a result, it caused an exponential increase in the paraffin/olefin ratio with carbon number but did not lead to deviation from Anderson–Schulz–Flory (ASF) distribution. In the case of model catalyst $Co/SiO_2$, $\alpha$-olefins were mainly re-included into the chain growth process, causing an increase in the growth probability. To a lesser extent, hydrogenolysis proceeded, and as a result of which, the number of carbons atoms in long hydrocarbons was reduced due to the consecutive methane elimination. The total effect of this process was expressed in sigmoid product distributions with a high selectivity to middle distillates.

Jam S. et al. [22] carried out a two-stage FTS and observed an increase in $C_{7+}$ n-paraffins content. According to the authors, it was a result of re-adsorption of $C_2$–$C_6$ $\alpha$-olefins with their subsequent inclusion in the growing chain. At the first stage of a fixed-bed reactor, an iron catalyst active in the synthesis of light olefins was used. The experiments were performed at a temperature of 320 °C with a ratio of $H_2/CO = 2$. At the second stage, a ruthenium-promoted cobalt catalyst was used, and the experiments were performed at a temperature of 220 °C with a ratio of $H_2/CO = 2$. A comparative analysis of the results obtained for each catalyst separately and during two-stage FTS allowed to establish that the complete disappearance of $C_2$–$C_6$ $\alpha$-olefins is explained by an increase in $C_{7+}$ n-paraffins content and not from their direct hydrogenation to the corresponding n-paraffins.

It was shown in [23,24] that in hydrocarbons obtained by FTS in the presence of both Co and Fe catalysts, methyl-branched hydrocarbons were identified, but ethyl-branched and dimethyl-branched hydrocarbons were not observed. The total amount of branched alkanes is about 5% in the presence of Co catalyst, while in the presence of Fe, it is about 25%. In order to explain the results, the authors proposed the alkylidene mechanism of isomerization. According to this mechanism, branched hydrocarbons are produced by re-adsorption of olefins [19]. The alkylidene mechanism predicts that (1) there will be «kink» corresponding to $C_2$ on the MWD since the ethylidene is more active than alkylidene in combination with the monomer MCH; (2) the branched hydrocarbons are monomethyl-branched with little amount of ethyl or multimethyl-branched hydrocarbons; (3) the formation of branched hydrocarbons will not obey ASF kinetics or the growth probability that branched hydrocarbons will be larger than linear chain hydrocarbons. The experiments with the deuterium have shown that the formation of 2-alkenes is different from the formation of 1-alkenes.

Therefore, the contribution of secondary transformations of hydrocarbons in the total composition of FTS products in the presence of Co catalysts is a promising direction to create catalytic systems for the selective production of narrow hydrocarbons' fractions—for instance, direct formation diesel fraction from CO and $H_2$ without additional stages of hydrotreatment requiring high energy costs. Zeolites are the most used catalysts for their components which are active in the transformations of hydrocarbons.

## 3. Zeolites Properties Responsible for Their Role in Bifunctional Catalysts

Zeolites belong to synthetic inorganic materials—porous crystals, which are aqueous aluminosilicates, from which water is removed without destroying the silicon–oxygen framework [25–30]. In this case, a regular porous system is formed in the crystal that consists of cavities and channels. The geometry of intra-crystalline pores—cavities and channels—determines the properties of zeolites and, consequently, the fields of application. On the surface of zeolites, there are OH-groups due to which zeolites have acidity, being solid acids. Zeolites have two types of reactive acid sites: Bronsted and Lewis acid sites. The dehydration of zeolites at temperatures above 450–500 °C leads to the removal of hydroxyl groups and the appearance of Lewis acid sites, which are cations or tricoordinated

aluminum atoms, which are located in places with oxygen deficiency or in the locations of cations. If the calcination temperature does not exceed 450 °C, then in the presence of water molecules, Lewis acid sites can turn into Bronsted acid sites [30]. In this case, the water molecule coordinating the aluminum atom and the bridging hydroxyl group acquire the properties of a hydroxonium ion, which manifests in the formation of shortened hydrogen bonds such as adsorption sites with water molecules.

The presence of acid sites makes zeolites catalytically active. At the same time, the strength of acid sites can be controlled. It depends both on the type of zeolite and its silica to alumina ratio (SAR) and the availability of acid sites to reactant molecules. Therefore, the acidity of zeolites can be controlled by means of (1) choosing the type of zeolite; (2) changing its SAR; (3) developing additional porosity that makes the acid sites available; and (4) introducing additional cations.

The adsorption of reagents and products determines the way that molecules interact with zeolites and is one of the key stages of catalytic transformations [30,31]. Three types of interaction were established after studying the adsorption of alkanes on zeolites [32–35]: (1) with zeolite lattice by dispersion Van der Waals forces, (2) with acid sites or cations and (3) intermolecular interactions of absorbed molecules. The strongest interaction is with the zeolite lattice [32]. The contribution of intermolecular interaction increases with increasing coverage degree. Despite the low contribution to the heat of adsorption, directed interactions can lead to the polarization of sorbed hydrocarbons [35]. The study of the adsorption of molecules that cannot penetrate into the pores of the zeolite suggests that catalytically active silanol sites (and for some zeolites, Lewis acid sites) also occur on the external surface of zeolites [25,36,37]. For example, in the HMOR zeolite, at least some of the strong Lewis acid sites are located on the external surface of the zeolite.

The use of zeolites as a component of catalysts to the synthesis of hydrocarbons from CO and $H_2$, in which water is the main by-product, requires taking into account the specific nature of interactions zeolites with water.

Water adsorption leads to a decrease in Si-Si distance and an increase in Al-Al and Si-Al distance. The maximum distance between oxygen atoms also increases. In the adsorbed state, the water molecule is distorted, and the distances O-H1 и O-H2 are equal to 0.980 Å and 0.962 Å, respectively, instead of 0.95096 Å, and become a possible source of proton acidity [38–40]. Thus, water adsorbed by zeolite is not an inert pore filler but has a significant effect on the adsorption and acidic properties of zeolite. Thus, it was shown in [41,42] that water can actively participate in transformations on the surface of zeolites in the H-form of even nonpolar hydrocarbons. According to the data obtained by MAS NMR in situ, it is assumed that this occurs due to the intensification of proton transport, which requires closely located acid sites.

The bimolecular or carbocation mechanism is the most used to explain the regularities of hydrocarbon transformations on the acid sites of zeolites. This mechanism was first proposed in 1934 [43] and is described in detail in [44–46], and less commonly used is the monomolecular or protolytic mechanism first published in 1984 [47,48]. Transformations by the bimolecular mechanism are characterized by the presence of an induction period and by the monomolecular one in its absence. The main products of the monomolecular mechanism are paraffins, including methane and ethane, and the main products of the bimolecular mechanism are low-molecular hydrocarbons with predominance olefins (methane, ethane and ethylene are not formed) [45,49]. In work [50], the possibility of n-hexane cracking by bimolecular and oligomerization mechanisms is discussed. The main difference in the second one is the occurrence of alkylation reactions of olefins with each other resulting in the formation of products with a higher molecular weight than the initial ones.

Among the reactions occurring at the acid sites of zeolites, it is possible to distinguish double bond transfer, isomerization, condensation, oligomerization, alkylation, dealkylation, hydrogen spillover and cracking [25,28,29,51].

The ratio between the rates of these reactions depends on the process parameters, the structure of the hydrocarbon molecule and the structure of the zeolite. The availability

of acid sites for the adsorption of hydrocarbons plays a significant role in the activity of zeolites, and many studies have shown that this parameter is more important than the strength of acid sites.

Therefore, a number of secondary transformations of FTS hydrocarbons can occur at the acid sites of zeolite. It is assumed that the isomerization of olefins—the primary products of the synthesis of hydrocarbons from CO and H$_2$—can occur at the Bronsted acid sites of any strength, and oligomerization, cracking and hydrogen spillover can occur at strong acid sites [26,28]. The availability of zeolite acid sites for the re-adsorption of hydrocarbons is also of great importance [52,53]. Data about activity of zeolites in the temperature range of 170–260 °C is insignificant and scattered. Nevertheless, the composition of hydrocarbons obtained in the presence of zeolite-containing catalysts largely depends on zeolite properties [9].

## 4. The Effect of Zeolite in FTS Co Catalysts on the Composition of Products

Therefore, polymerization reactions of CH$_x$ monomers with the formation of paraffins and olefins, hydrogenation of CO to methane and the resulting olefins to paraffins, inclusion of olefins in the growing hydrocarbon chain and hydrogenolysis of hydrocarbons with the formation of methane can occur on the surface of FTS Co catalysts (Figure 2). Isomerization of hydrocarbons, cracking, oligomerization, alkylation, de-alkylation, condensation, aromatization, hydrogen spillover, double bond migration and alcohol dehydration can occur at the acid sites of zeolites. The combination of reactions at different active sites is a promising methodology to control selectivity of FTS. It will allow to deviate from the MWD corresponding to the polymerization mechanism for traditional catalyst systems with one type of active sites [20,54]. The use of zeolite as a component of FTS catalyst can make it possible to increase the selectivity of formation—for example, gasoline or diesel fraction components, olefins C$_2$–C$_4$ or heavy hydrocarbons with a narrow range of molecular weights.

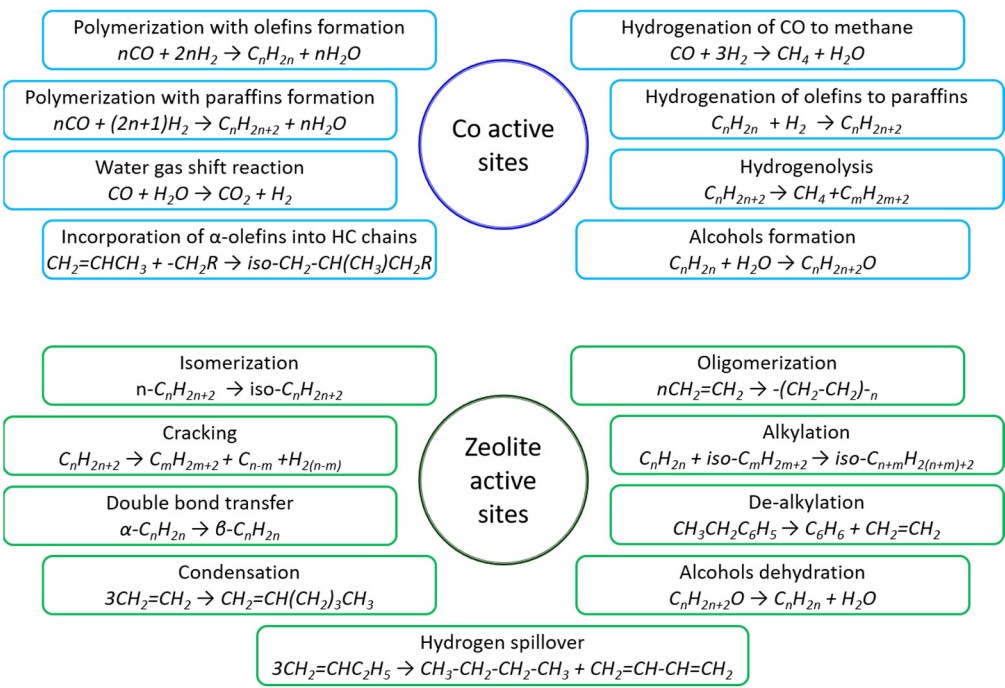

**Figure 2.** Possible reactions on the Co and zeolite active sites.

In addition, the composition of FTS products in the presence of bifunctional catalysts will be influenced by the location of the active sites relative to each other, including the distance between them and accessibility—in particular, the availability of acid sites for re-adsorption of FTS hydrocarbons for further transformations, i.e., the parameters of

the porous system such as catalysts [9,10]. Consequently, the pellets-forming method is an important factor (Figure 3). In the literature, besides the conventional impregnating catalysts, in which the metal active in FTS is applied directly to zeolite, core-shell catalyst, bimodal or catalyst with hierarchical pore distribution, as well as physical mixtures of FTS catalyst and zeolite, both granular and non-granular are discussed [55–57].

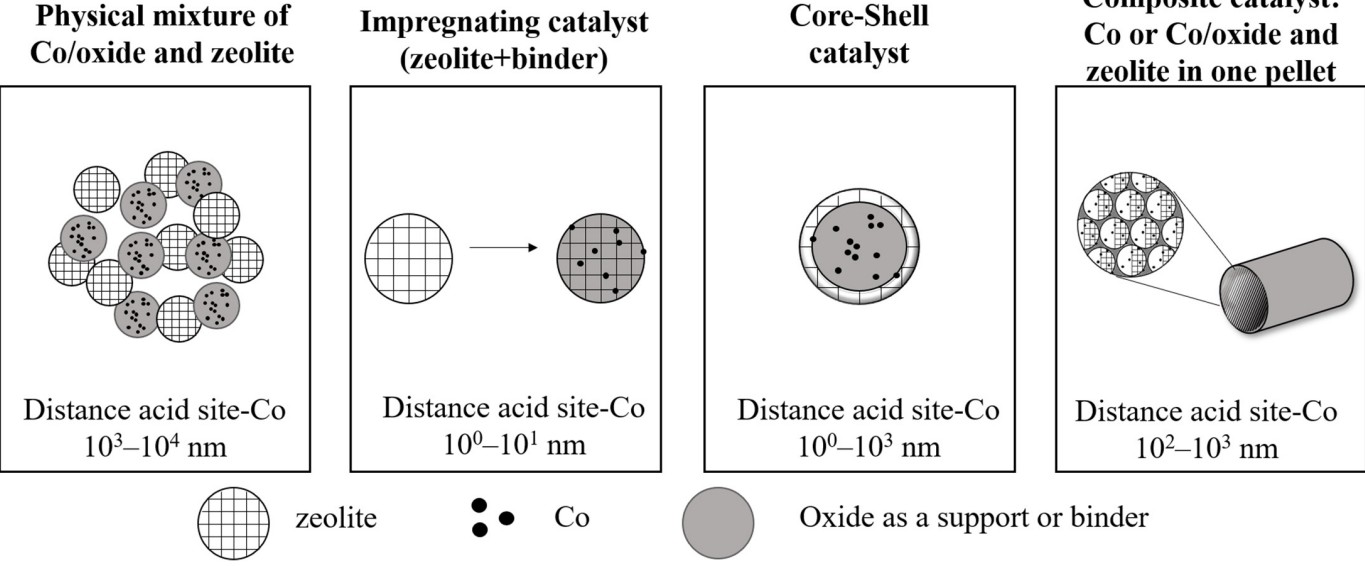

**Figure 3.** Different ways of a catalyst bed organization.

### 4.1. One-Step Production of Hydrocarbons Gasoline Fraction

The greatest number of publications devoted to the bifunctional FTS catalyst suggest the introduction of zeolite into the catalyst composition for the selective production of gasoline-range iso-paraffins. One of the discussed ways to organize active sites of different nature in one catalytic system is the core-shell catalyst [55]. It is constructed by applying a zeolite membrane (shell) on the surface of a preformed catalytic pellet (core). The syngas firstly penetrates through the shell membrane, reacts on the metal sites active in the FTS and forms hydrocarbons, which, passing through the shell, undergo transformations on the acid sites of zeolite.

Yang G. et al. [58] suggest two methods for capsule catalyst preparation FTS 25%Co/SiO2 in zeolite HZSM-5: hydrothermal synthesis and the physically adhesive method. The thickness of the zeolite shell obtained by hydrothermal synthesis was 24 $\mu$m. Catalyst pellets with size range of 0.85–1.70 mm were tested in a fixed-bed reactor under pressure of 1 MPa and temperature of 280 °C with the syngas ratio of $H_2/CO = 2$ and a rate of 2.24 L/(g·h). In the presence of both catalysts, the selectivity of light iso-paraffin was increased from 36% obtained by physically mixed method $Co/SiO_2$ and HZSM-5 to 44–49%, concurrently decreasing the formation of heavy hydrocarbons. An increase in the zeolite content from 25 to 60 wt.% contributed to an additional increase in the selectivity of the formation of light iso-paraffins up to 61% in the presence of a catalyst prepared by the physically adhesive method. The authors explain it through the high porosity of the zeolite layer, which reduces diffusion limitations.

In works [59–61], Co and $SiO_2$ catalysts were also used as a core, and zeolite HZSM-5 was used as a shell to obtain $C_5$–$C_{11}$ hydrocarbons directly from CO and $H_2$. The shell was formed through hydrothermal synthesis. However, this method differed from other and previous methods, as described in [58]. The FTS was conducted in the temperature range of 200–260 °C and pressure of 1 or 2 MPa with molar ratio syngas equal 2. Cobalt loading in the initial catalyst (before the creation of a zeolite shell) was 10–25 wt.%. Regardless of the method preparation and of the presence of promoter, all core-shell catalysts were characterized as significantly higher in gasoline selectivity (up to 70%) and produced more iso-paraffins (up to 25%) compared to a conventional $Co/SiO_2$ catalyst, as authors [59–61]

explained the impossibility of FTS hydrocarbons desorption from the catalyst surface without contact with the acid sites of the zeolite. In addition, there was a decrease in the amount of carbon deposits on the catalyst surface containing zeolite and their high stability.

Lin Q. et al. [62] suggested core–shell catalysts, in which the core is a Pd-promoted 10%Co/SiO$_2$ catalyst and the shell isa zeolite HZSM-5 for direct synthesis of C$_5$–C$_{11}$ iso-paraffins from syngas via FTS, which was conducted at a temperature of 260 °C and 1.0 MPa, H$_2$/CO = 2. It was demonstrated that, in the synthesis, Co species in the Pd-promoted catalyst represented the cobalt silicate obtained during the hydrothermal synthesis, but in the un-promoted catalyst, it was Co$_3$O$_4$. Despite Pd-promoted capsule catalyst being more selective in the formation of C$_5$–C$_{11}$ iso-paraffins (22%) with molar ratio of iso-paraffins to n-paraffins up to 1,03 CO, conversion in the presence of the Pd-promoted catalyst did not exceed 31%.

In work [63], the core-shell catalyst was described: the core was skeletal Co, and the shell was zeolite HZSM-5. The FTS was conducted in a fixed-bed reactor under pressure of 2 MPa and temperature of 250 °C with the syngas H$_2$/CO = 2. The activity of this catalyst was lower than the activity of the physically mixed skeletal Co-HZSM-5 catalyst, but the selectivity of C$_5$–C$_{11}$ reached 79%, and high molecular weight hydrocarbons were completely cracked.

The FTS selectivity to gasoline-range hydrocarbons was studied in [64]. The catalyst was obtained by coating controlled-thickness ZSM-5 film on the surface of Co-Al$_2$O$_3$/monolith substrates. The monolith serves as an efficient heat transfer structure, and the ZSM-5 layer functions as a cracking and isomerization catalyst. Obtained liquid hydrocarbons contained gasoline yield higher than 65 wt.% and selectivity to isomers and olefins greater than 70 wt.%. These parameters were obtained during FTS at a temperature of 230 °C and under pressure of 1.2 MPa, and 79% CO conversion was reached.

Therefore, zeolite HZSM-5 is most commonly used as a shell capsule catalyst for the direct production of gasoline hydrocarbons from CO and H$_2$. However, other examples can be found. In work [65], catalyst 10%Co/SiO$_2$ was immersed in a mixture of zeolites MOR and ZSM-5 obtained through the solvent-free synthesis method: boehmite, Na$_2$SiO$_3$·9H$_2$O and Co/SiO$_2$ were sequentially mixed, followed by heating in an autoclave at 200 °C. In work [66], pellets of 7%Co/Al$_2$O$_3$ were capsulated in a H–β zeolite shell through a hydrothermal synthesis method. The weight of the catalyst was 0.5 g, and the FTS reaction was conducted in a fixed-bed reactor and tested at a temperature of 265 °C and 1 MPa. Li X. et al. report that the formation of heavy paraffins was completely suppressed, and iso-paraffins C$_4$–C$_7$ were the main products. However, the molar ratio of iso-paraffin to n-paraffin of the products obtained from the encapsulated catalyst increased about 64% more than the molar ratio of the products obtained from physical-mixed components.

With all the obvious advantages of core-shell catalysts in increasing the selectivity of gasoline-fraction-formation iso-paraffins, their specific activity is insufficient, especially in industrial conditions when to ensure cost-effectiveness requires the use of high feedstocks rates since such an organization of the catalyst expects strong transport difficulties in the diffusion of products and reactants through the zeolite shell. In work [67], different catalyst configurations such as impregnating catalysts Co/SiO$_2$ and H-ZSM-5, encapsulated catalyst HZSM-5/Co/SiO$_2$ and physical mixture of Co/SiO$_2$ and HZSM-5 were compared. The authors concluded that for the direct synthesis of C$_5$–C$_{11}$ iso-paraffins, the impregnating catalyst based on the mesoporous zeolite HZSM-5 is preferred.

An impregnating FTS catalyst containing zeolite is the most common with developers of bifunctional catalysts since in such systems, active sites of two types are in close contact with each other, facilitating re-adsorption of FTS products and, therefore, affecting the composition of the products [68].

It was demonstrated in [69] that by using X-Ray diffraction analyses for catalysts prepared by impregnation of zeolites H-USY, H-Beta, H-Mordenite and H-ZSM-5 with SAR of 13, 7, 27 and 16, respectively, and containing 10 wt.%, Co characteristics for each zeolite structure were maintained. The average Co particle size was increased in the range

Co/H-Beta < Co/H-USY < Co/HMordenite < Co/H-ZSM-5 and depended on the zeolite structure. The highest selectivity of $C_5$–$C_9$ hydrocarbons formation 55–57% was prepared in the presence of H-Mordenite and H-Beta catalysts. Zola A.S. et al. suggested that the composition of the hydrocarbons also depended on secondary porosity, which makes zeolite sites more accessible for re-adsorption of FTS products.

Baranak M. et al. [70] used low acidic ZSM-5 for its lower activity in respect to zeolite-acid-catalyzed reactions, i.e., hydrocracking, which may lead to high selectivity for gasoline range components. Zeolite ZSM-5 with SAR 280 was impregnated with iron nitrate solution $(Fe(NO_3)_3 \cdot 9H_2O)$, and a hybrid catalyst was prepared through the physical mixture method of the base iron and ZSM-5. In addition, the impregnating catalyst was prepared based on zeolite ZSM-5, previously dealuminated by oxalic acid solution. The FTS reactions were carried out at a temperature of 280 °C, 1,9 MPa with syngas molar ratio of $H_2/CO = 2$ and with gas hourly space velocity (GHSV) 750 $h^{-1}$. CO conversions were higher than 40% for all zeolite containing catalysts. The selectivity of $C_5$–$C_{11}$ hydrocarbon fraction of catalysts prepared through impregnation was 50–74%, and for catalysts prepared through the physical mixture method, it was 45%. In the case of the impregnating catalyst, the composition of the FTS products did not observe heavy hydrocarbons. About 2 wt.% heavy hydrocarbons was measured in the case of catalysts prepared through the physical mixture method.

It was proposed in [71] to increase the activity and selectivity of the Co/HZSM-5 catalyst obtained through the impregnation method in the formation of gasoline range hydrocarbons by introducing promoters Ru and Ni. The FTS was carried out under the pressure of 2 MPa, temperature of 235–300 °C, $H_2/CO = 2$ with GHSV 1500 $h^{-1}$. The CO conversion was increased with the promotion of both metals and the content of iso-paraffin in the gasoline range (38% at 300 °C)—in the case of the Ni promoter. In addition, the highest selectivity of the $C_5$–$C_{12}$ hydrocarbons formation in the presence of a catalyst containing Ni was obtained at a lower synthesis temperature—60% at 251 °C. However, the catalyst containing Ni had the shortest time on stream. Wang S. et al. found that it depends on the maximum amount of active cobalt sites in promoted Ni catalysts.

Kang S. et al. [72] used promoted Ru, Pt or La Co/ZSM-5 catalyst (Si/Al = 25) obtained through the impregnation method for the direct production of gasoline-range hydrocarbons from syngas. The promoted catalysts were prepared through co-impregnation of the nitrate (Ru, Pt or La) and chloride. The FTS was carried out in a fixed-bed reactor under a temperature of 240–260 °C, 2.0 MPa and 3000 mL/$g_{cat}$/h with $H_2/CO = 2$. Increasing the activity and selectivity of the catalyst containing Pt through co-impregnation of the nitrate in the production of gasoline-range hydrocarbons is explained with the high reducibility of cobalt particles. It was achieved due to the presence of weak acid sites, a large pore volume with a large diameter of pores and small sizes of Co particles. The selectivity of the formation of $C_5$–$C_9$ hydrocarbons was 26 mol.% at a temperature of 260 °C.

It was shown in [73] that the promoted 0.1 wt.% Ru catalyst 11 wt.% Co/ZSM-5 (Si/Al = 25) demonstrates an increase in the yield of gasoline hydrocarbons of up to 72%. The authors explained the achieved effect through the influence of polyethylene glycol added to the impregnation solution, which contributed to the deposition of $Co_3O_4$ particles mainly on the external surface of ZSM-5. Additionally, bimetallic Co-Ru particles were formed, which weakened the H–catalyst surface interaction. Increasing the SAR of zeolite ZSM-5 up to 80 in the composition of catalyst Ru-Co/ZSM-5 prepared through the impregnation method led to a decrease in the selectivity of the formation of gasoline-range hydrocarbons to 56%, despite a twofold increase in activity [74]. The authors explained the result through a decrease in the number of Bronsted acid sites with an increase in the zeolite modules.

In work [75], the effect of Co localization, structure and acidity of zeolite on the formation of iso-paraffins from CO and $H_2$ in the presence of Pt-promoted catalysts containing zeolites ZSM-5, MOR and Beta was investigated. The catalysts were prepared through the impregnation method of zeolite and through mechanical mixing Co/SiO$_2$ FTS catalyst with

zeolite. The FTS was carried out under the following parameters: 2 MPa, 250 °C, molar ratio $H_2/CO = 2$ and 34 L/($g_{Co}$·h). It was shown that the increase in the pore size and open nature of the zeolite structure from ZSM-5 to Beta led to an increase in the fraction of Co located inside the pores of the catalysts prepared through impregnation. The most active in the formation of short-chain iso-paraffins were catalytic systems containing zeolite ZSM-5. In their presence, hydrocarbons containing up to 20% of $C_5$–$C_{12}$ iso-paraffins were formed. It was found that the selectivity to short-chain iso-paraffins is affected mainly by the zeolite acidity, and the selectivity to long-chain branched hydrocarbons mostly depends on steric effects.

In work [76], the cobalt in the amount of 7.5, 10, 15 and 20 wt.% was loaded on the nanoscale Beta zeolite ($SiO_2/Al_2O_3 = 50$) through impregnation. The FTS was carried out under the following parameters: 1 MPa, 220 °C and syngas molar ratio $H_2/CO = 2$. The formed $C_5$–$C_{20}$ hydrocarbons were characterized by an average carbon number in the range $C_8$–$C_{10}$, and the largest number of isomers (32%) was obtained in the presence of a catalyst containing 15% Co. It is particularly noted that all the studied catalysts were already active in isomerization at 220 °C due to the small size of zeolite crystallites. It is assumed that finely dispersed cobalt particles strongly interact with the external surface of the zeolite, causing partial poisoning of acid sites and reducing the proportion of iso-paraffins.

Less common zeolites—for example, cancrinite (CAN)—are also used as a Co-catalyst support for FTS with increased selectivity to produce gasoline-range hydrocarbons [77]. CAN's structure is characterized as a large uni-directional channel that, according to the authors, facilitates the diffusion of FTS products to the acid sites of the zeolite. In addition, the highest selectivity of the formation of target products (78%) was obtained in the presence of a catalyst based on the Na-form of cancrinite, which the authors explain by the presence of weak acid sites less active in deep cracking.

In recent years, a popular direction for optimizing the properties of bifunctional catalysts obtained by impregnation is the use of supports with a hierarchical or bimodal pore structure since they can reduce or even remove diffusion limitations. In work [8], mesostructures in HZSM-5 zeolite crystallites (Si/Al = 40) were prepared through subsequent base and acid treatments. According to the authors, the first treatment contributed the formation of large pores; the second removed non-framework aluminum; and all this combined led to an increase in the activity of the FTS catalyst 10%Co/HZSM-5 for the direct production of gasoline-range hydrocarbons. The FTS reactions parameters were the following: 1.5 MPa, 220–240 °C and syngas molar ratio $H_2/CO = 2$. It was found that hydrocracking of primary FTS hydrocarbons increased the selectivity of $C_5$–$C_{11}$, and the highest parameter (59%) was obtained in the presence of a zeolite-based catalyst treated with both base and acid. On the other hand, the strong Co–zeolite interaction as revealed by TPR($H_2$) resulted in the stabilization of lower coordinated Co sites and in a higher selectivity of methane formation. The study of n-hexane conversion reactions in the presence of a catalyst based on a support with a mesostructure suggested that increased methane formation is associated with high activity of the catalyst in hydrogenation and hydrogenolysis reactions at such coordinatively unsaturated Co sites.

In work [78], HZSM-5 zeolite with combined micropores and mesopores structure was prepared using propylene oxide as a subsequent support of Co-catalyst FTS. Indeed, mesopore volume increased 2.5 times, while the size of $Co_3O_4$ crystallites decreased from 16.1 to 12.7 nm, and acidity increased from 2.02 to 2.26 mmol/g. The FTS catalysts with the Co loading amount of 10 wt.% were prepared through the impregnation method. It is assumed that the presence of mesopores contributed to an increase in CO conversion (from 76 to 79%), selectivity of $C_5$–$C_{11}$ hydrocarbons formation (from 56.6 to 65.4) and iso-paraffin/n-paraffin ratio in $C_{4+}$ hydrocarbons (from 0.5 to 0.82) in FTS at 240 °C and 1 MPa.

In work [79], the increased selectivity of 10%Co/MFI catalyst (HZSM-5 with module 40) in the formation of branched $C_5$–$C_{11}$ hydrocarbons is also explained by an additional mesoporosity, which was prepared using nanosponge as a catalyst support. Nanosponge

consists of a disordered network of 2.5 nm-thick MFI zeolite nanolayers and a narrow distribution of mesopore with a maximum of 4 nm. The Co particles supported on the obtained structure were characterized as narrow particle size distribution with a maximum of 4 nm and resistant to agglomeration. In the presence of such catalyst under the following FTS parameters 220 °C, 2 MPa, $H_2/CO = 2$ and 2.4 L/(h·g), the selectivity of formation $C_5$–$C_{11}$ hydrocarbons was 75%, and the proportion of isomers was 28%. Moreover, the catalyst was characterized as having high stability over 100 h of reaction time. The authors explained the results with a short diffusion distance in thin zeolite frameworks.

It was proposed in [80] to use mesoporous H-ZSM-5-carbon composites obtained by deposition of pyrolytic carbon on just synthesized zeolite as a support for the preparation of Co (17–24%) FTS catalyst. This made it possible to increase the selectivity in the formation of $C_5$–$C_{11}$ hydrocarbons to 57%, while the selectivity in the formation of $C_1$–$C_2$ hydrocarbons slightly decreased. The authors of [80] explain the results with the lower Co–support interaction (by XPS analyses), which leads to the high cobalt reduction degree. Moreover, the partial deactivation of the Bronsted acid sites through pyrolytic carbon deposition allows the modification of the zeolite acidity and, therefore, the composition of products. The FTS reactions were carried out under the pressure of 2 MPa at a temperature of 240 °C and $H_2/CO = 2$.

In [81], the possibility of obtaining from syngas of gasoline-range hydrocarbons with selectivity of up to 70% and a ratio of iso-paraffins/n-paraffins of 2.3 in the presence of Co particles supported on mesoporous zeolite H-ZSM-5 was investigated. The authors concluded that the Bronsted acidity resulted in hydrocracking/isomerization of the heavy hydrocarbons formed on Co nanoparticles, while the mesoporosity contributed to suppressing the formation of light hydrocarbons. In addition, using n-hexadecane as a model compound, it was determined that both the acidity and mesoporosity played key roles to control the hydrocracking reactions and contributed to the improved selectivity of hydrocarbon ($C_5$–$C_{11}$) formation.

Therefore, in order to direct the production of gasoline-range hydrocarbons, zeolite H-ZSM-5 is more often used. Moreover, regardless of the organization method of the catalyst layer, generally, the FTS carried out at a temperature of 220–260 °C and pressure of 1–2 MPa.

### 4.2. Selective Production of Isoparaffins

Along with the studies dedicated to the direct production of gasoline-range hydrocarbons, it can be noted that studies were also aimed at increasing the yield of iso-products regardless of their molecular weight. In work [82], iso-paraffins were the main product of FTS with selectivity up to 52.3% due to the optimized hydrocracking and isomerization provided by the hierarchical structure of zeolite and the high ratio of Bronsted to Lewis acidity. The hierarchical zeolite Y with intracrystalline mesopores was prepared through subsequent base and acid treatments. After that, 10 wt.% Co was applied to it. The authors noted that the conversion and C5+ selectivity in the presence of catalysts based on hierarchical zeolite Y was higher than in the case of untreated zeolite. The FTS was carried out in a flowing fixed-bed reactor at 260 °C under pressure of 1.0 MPa and $H_2/CO = 2$.

In work [83], the increased selectivity in the isomerization reaction (up to 21%) is also explained by the hierarchical micro-meso-macroporous structure of Beta zeolite, which is characterized by the presence of strong acid sites and reduced diffusion limitations due to macropores. Mesopores were formed during the preparation of zeolite through crystallization, followed by steam treatment to form nanoscale crystals, which contributed to obtaining a hierarchical pore distribution. Cobalt was impregnated in an amount of 15 wt.%. FTS was carried out at 225 °C and 2 MPa, and a molar ratio of $H_2/CO = 2$ was supplied at a rate of 4.8 L/(g·h).

Flores C. et al. [84] note that high selectivity to production branched hydrocarbons obtained in the presence Co catalyst based on mesoporous ZSM-5 zeolite was attributed the reduced diffusion limitations. The use of carbon nanotubes impregnated with cobalt

as a template in the synthesis of zeolite changed the morphology and structure of zeolite: the mesopores volume of such zeolite increased by 3–4 times. This facilitated the transport of isomers and the prevention of their cracking, which made it possible to obtain $C_{5+}$ hydrocarbons containing 63 wt.% iso-paraffins. Moreover, such catalysts were 5–8 times more active in FTS compared to catalysts prepared through conventional impregnation. The FTS reaction was conducted at 250 °C and 2 MPa with a molar ratio of $H_2/CO = 2$ and a rate of 20–70 L/($g_{Co}$·h).

In [85], cobalt distribution between the external surface and micropores of the large pore HBeta zeolite were controlled by choosing the sequence of impregnation and ion exchange method. In this connection, the strength and number of Bronsted and Lewis active sites were changed. The FTS reaction was conducted at 250 °C and 2 MPa with molar ratio$H_2/CO = 2$ and a rate of 66 L/($g_{cat}$·h). It was found that higher reaction rates were observed over the catalysts, which did not contain cobalt ions in the cation sites of the zeolite. Lower methane selectivity and high fraction of isomerized products (67 wt.% in $C_{9+}$) are observed in the presence of catalysts, in which cobalt species are located on the zeolite external surface and acid sites inside the HBeta zeolite micropores. The authors also note that the Na-form of zeolite HBeta contributed to an increase in the content of the long chain.

The effects of the localization of Co species and zeolites acidity ZSM-5, MOR and BEA (Si/Al 13, 8 and 9, respectively) on the performance of Pt-promoted catalysts for the direct synthesis of iso-paraffins from syngas were investigated in [75]. The catalysts were prepared through the impregnation method of the zeolite, using salts Co and Pt and by the mechanical mixing of Co-Pt catalyst with zeolite. The Co and Pt content were 20 and 0.1 wt.%, and the ratio Co:zeolite in the mechanical mixtures was 4:1. It was shown that the increase in the pore size and open nature of the zeolite structure from ZSM-5 to BEA resulted in a higher fraction of Co located inside the pores of the catalysts prepared through impregnation. The hydrocracking and isomerization rates were dependent for both the zeolite acidity and its pore structure. The selectivity in the $C_5$–$C_{12}$ hydrocarbon range was mostly affected by the number of strong acid sites, and a higher concentration of iso-paraffins was observed in the presence of ZSM-5. The authors concluded that activity in the formation short-chain iso-paraffins was affected mainly by the zeolite acidity, and the selectivity to long-chain branched hydrocarbons mostly depended on spatial effects such as pore size, the open nature of the zeolite structure as well as the localization of Co species. Moreover, the selectivity of the long-chain iso-paraffins was contributed by close contact between Co metallic species and Bronsted acid sites.

Therefore, for the successful isomerization of hydrocarbons under FTS conditions, the concurrent presence of both strong Bronsted acid sites and meso- and macropores is necessary, and the preferred location of cobalt particles is the external surface of zeolites.

*4.3. Single-Stage Production of Diesel Fuel Components*

There are significantly fewer works devoted to the selective production of diesel fuel components in the presence of bifunctional FTS catalysts than those devoted to the direct production of gasoline-range iso-paraffins. It should be noted that the main way to obtain such catalysts is to apply the active component to the surface of a zeolite-containing support since it is difficult to control the degree of cracking in core-shell catalysts. Almost every hydrocarbon molecule passing through the zeolite shell will undergo cracking at each contact with the acid site of zeolite.

The authors in [86,87] suppose that the effective control of hydrogenolysis using mesoporous zeolite nanoparticles can enhance the diesel fuel selectivity while keeping methane selectivity low. In their work, 15 wt.% Co was applied to mesoporous zeolite Y through the melt infiltration and impregnation methods. The melt infiltration method led to the formation of Co particles with a narrower size distribution in comparison with impregnation. The catalyst obtained through the melt infiltration method was characterized by a higher selectivity of $C_{10}$–$C_{20}$ hydrocarbon formation. The zeolite Y in H- and Na-forms

was chosen due to having a weaker Bronsted acidity than ZSM-5. FTS was performed under a pressure of 2 MPa at 230 °C with $H_2/CO = 1:1$ and 20 mL/min. The most selective of diesel fraction hydrocarbons (60%) was a catalyst based on Na-type mesoporous Y-supported Co catalyst with Co particles with sizes of 8.4 nm and mesopores diameter of 15 nm obtained through the melt infiltration method. The addition of Mn led to an increase in the diesel fraction selectivity up to 65% by suppressing the formations of $CH_4$ and lighter hydrocarbons. The results of the model reaction of hexadecane hydrogenolysis in the presence of the obtained catalysts confirm that the hydrogenolysis of heavy hydrocarbons occurs under FTS conditions. The authors propose that the narrow Co particles distribution favors a decrease in the contribution of hydrogenolysis and an increase in the selectivity of the $C_{10}$–$C_{20}$ formation.

In works [88,89], the effect of the hierarchical H-ZSM-5 prepared through alkali treatment on the selectivity of an impregnated Co catalyst based on the zeolite (Co loading is 15 wt.%) was investigated. The FTS was conducted at 210 °C and 2 MPa with a molar ratio of $H_2/CO = 2$ and 1200 $h^{-1}$. An increase in the proportion of mesopores in zeolite was achieved with varying alkali concentrations. It was found that the catalyst treated with 0,1 M NaOH is optimal for $C_{12}$–$C_{18}$ selectivity, which reached 35%. The authors suggest that this factor is due to the bimodal porous structure which promotes the easier diffusion behavior of products combining with acid sites of moderate strength to reduce the depth of secondary reactions. It led to an increase in the ratio of iso-paraffins and olefins of $C_5$–$C_{18}$, which indicated an increase in the availability of acid sites for re-adsorption primarily formed hydrocarbons. Moreover, the authors suggest that alkali treatment produces $\alpha$-$SiO_2$ phase, which leads to an increase in cobalt-support interactions, which reduces catalyst activity in FTS and higher selectivity of methane formation.

In works [90,91], a composite catalyst to obtain a low-cold diesel fraction from CO and $H_2$ is proposed. Such a catalyst is a mixture of 20Co-1$Al_2O_3$/$SiO_2$ catalyst active in long-chain hydrocarbons synthesis and zeolite HZSM-5 with different modules, formed into pellets using a binder (boehmite). Before testing, the pellets were crushed to 1–2 mm and mixed with quartz. After reduction with hydrogen at 400 °C, catalysts were tested in FTS at 2 MPa, 180–250 °C with a syngas ratio $H_2/CO = 2$ and 1000 $h^{-1}$. The highest selectivity in the formation of $C_{5+}$ (67–75%) was observed at 240 °C, while they contained 29–41 wt.% $C_{11}$–$C_{18}$ of hydrocarbons. The presence of zeolite in the composition of the composite catalyst contributed to an increase in the ratio of iso-paraffins and olefins in this fraction, while the ratio of $C_{19+}$ hydrocarbons in fraction $C_{5+}$ decreased from 47 to 7 wt.%. In most cases, this led to a decrease in turbidity temperatures and loss of fluidity. The authors concluded that $C_{5+}$ composition obtained directly from Co and $H_2$ was determined by the acidity of zeolite HZSM-5.

However, from our point of view, the direct production of synthetic oil (which is often called a mixture liquid of hydrocarbons obtained from CO and $H_2$ and does not contain components with a boiling temperature above 350 °C) as a monoproduct is a more promising direction of FTS [92] since it can be used as a feedstock for both the production of motor fuel components and in petrochemistry. In addition, it is possible to implement a modular technology for converting natural and associated gases from low-yield fields to synthetic oil as a final product for pipeline transportation in blend with mineral oil [93].

### 4.4. Direct Production of Synthetic Oil

The main type of catalysts offered in such implementation of FTS are impregnating catalysts, in which zeolite is used as a component of support. In [94], commercially viable bifunctional catalysts offered by Chevron (GCC™) are described. As an active metal, these catalysts contain 7.5 wt.% of Co promoted by 0.19 wt.% of Ru and the support based on zeolites ZSM-5 and ZSM-12. The catalysts were tested in a wide range of FTS conditions (205–235 °C, 0.5–3.0 MPa, $H_2/CO = 1–2$). Both catalysts were active in the formation of synthetic oil that does not contain high-molecular-weight hydrocarbons and alcohols with high selectivity 72–78%. The ZSM-12-containing catalyst was active in the formation of

synthetic oil enriched with hydrocarbons of the diesel fraction ($C_9$–$C_{19}$). The reason for this is that the straight ZSM-12 channels may transport heavier olefins to hydrogenation sites. In the presence of ZSM-5, the catalyst produced a lighter mixture of hydrocarbons while the yield of ($C_1$–$C_4$) increased. The authors explained the absence of alcohols in the synthesis products with the activity of zeolites in H-form in their dehydration. In addition, it was emphasized that the presence of zeolite as a component of the catalyst makes it possible to increase the performance of the catalyst up to 200 $g_{C5+}$/(kg$_{cat}$·h) with an increase in the proportion of hydrogen in the initial synthesis gas by two times.

In studies [93,95,96], a cobalt-zeolite catalyst containing an additive of aluminum metal powder or exfoliated graphite to increase thermal stability under exothermic FTS conditions and activate the direct production of synthetic oil from CO and $H_2$ is described. The Co loading is 20 wt.% while the optimal zeolite component is HBeta zeolite with module 38. The catalyst was tested under the following FTS conditions: 2MPa and at the temperature range of 170–260 °C, with syngas molar ratio $H_2$/CO = 2 and 1000–5000 h$^{-1}$. The authors note that due to the presence of a heat-conducting additive and zeolite in the composition of the catalyst, its performance reaches almost 600 $g_{C5+}$/(kg$_{cat}$·h) since the first component provides intensive heat transfer in the catalyst pellets, and the second one provides mass transfer by reducing the average molecular weight of the formed $C_{5+}$ hydrocarbons, in which the content of $C_{19+}$ does not exceed 2 wt.%.

The same group of works includes studies on the direct production of motor fuel from CO and $H_2$ without increased selectivity in the formation of individual fractions. The advantage of ultrasonic cobalt impregnation of nanoscale zeolite ZSM-5 over conventional impregnation using water or alcohol as a solvent to the selective production of motor fuels was demonstrated [97]. Nanoscale ZSM-5 was prepared through a steam crystallization method, and 15 wt.% Co was applied from a suspension of $Co_3O_4$. In the presence of the catalyst obtained through the ultrasonic impregnation method under 1 MPa and 230 °C, $C_{5+}$ hydrocarbons were formed free of $C_{20+}$, while the selectivity of motor fuel formation components was 67.8%. The authors explain these results with the strong acidity of nanoscale zeolite and close location between Co and acid sites.

The effect of mesopores on the Co dispersion, structural properties of zeolite ZSM-5, its crystallinity, the density of acid sites and the resulting Co/ZSM-5 catalyst in FTS was studied in [98]. Mesopores were introduced into the ZSM-5 crystals through two methods: alkali treatment and controlling the degree of crystallization. Regardless of the preparation method, the presence of mesopores in ZSM-5 increased the dispersion of Co nanoparticles and the acidity of zeolite, while the distance between the active sites of Co and zeolite was reduced. The FTS was conducted at 2MPa and 240 °C, and syngas with molar ratio $H_2$/CO = 2 was supplied at 4 and 10 L/(g$_{cat}$·h). The authors conclude that mesopores do not reduce diffusion limitations and do not contribute to the intensification of mass transfer of reactants and intermediate components. Their main advantage is to form well-dispersed Co active sites and regulated acid properties that facilitate the high conversion of CO and selectivity in the formation of liquid fuel fraction (including gasoline $C_5$–$C_{11}$ hydrocarbons with content of 35%).

### 4.5. Direct Production of Olefins

Olefins are an important source of feedstock for chemical and petrochemical industries—for example, the production of detergents and polymers. Direct synthesis of olefins from CO and $H_2$ in the presence of bifunctional catalysts continues to be of keen interest to the scientific community [99,100]. However, the well-known Fe-based catalysts of the Fischer–Tropsch to olefins (FTO) process operate under relatively severe conditions [101]. Co-based catalysts can be the optimal choice for a low-temperature process since they are characterized by their high activity and selectivity in the hydrocarbon's formation, lower activity to water gas shift and high activity [102]. Their high activity in olefin hydrogenation which is usually accompanied by low selectivity for olefins can be expected [103]. Nevertheless, it was possible to find only one study devoted to the selective production of olefins from CO and $H_2$ in the presence of

the Co-zeolite catalyst. In [104], it is proposed to use 10%Co/SiO$_2$ particles encapsulated in a hierarchal zeolite NaZSM-5 with subsequent applying of capsules to a heat-conducting SiC matrix for direct middle olefin (C$_5$–C$_{11}$) synthesis from CO and H$_2$. It is found that the high thermal conductivity of the SiC matrix benefits the high selectivity for long-chain hydrocarbons, and the hierarchical capsule structure regulates the product distribution. The FTS was conducted at a temperature of 300 °C with a molar ratio of H$_2$/CO = 2 and 1MPa. Indeed, the authors managed to increase the selectivity in the olefin's formation up to 32.3%, and their content in C$_5$–C$_{11}$ fraction was reached 49%. If the Na-form is replaced with H-form, it is possible to obtain a catalyst with increased selectivity in the iso-paraffins formation (27.3%).

In addition, in some studies devoted to FTS in the presence of Co-zeolite catalysts, there is also an increased yield of olefins. Thus, in the presence of a 10 wt.% Co-containing catalyst deposited on HZSM-5 zeolite with a hierarchal pore system, hydrocarbons C$_{5+}$ containing 30% of olefins were obtained in [78]. FTS was carried out at a temperature of 240 °C, 1 MPa with a molar ratio of H$_2$/CO = 2. The authors note that the formation of C$_{18+}$ was completely suppressed due to organized cracking-source of olefins C$_5$–C$_{11}$, which was facilitated by a hierarchal pore system.

## 5. Possibilities for the Product Composition Control

A review of the literature in the field of FTS in the presence of Co-zeolite catalysts has shown that the synthesis conditions do not have a decisive effect on the composition of hydrocarbons C$_{5+}$: gasoline fraction hydrocarbons, diesel and synthetic oil enriched with iso-paraffins and olefins. All the above-listed products are obtained under close conditions: at 1–2 MPa in a relatively narrow temperature range of 210–280 °C from a mixture of H$_2$ and CO with a molar ratio of 2 (Table 1). The greatest influence of the synthesis conditions is probably the contact time; however, researchers use different expressions of the flow rate, which are difficult to compare. It can be noted that in [85], the influence of the synthesis gas velocity in the range of 1000–6000 h$^{-1}$ was studied, which showed that a decrease in the contact time of the synthesis gas with the Co-zeolite catalyst led to a decrease in the content of hydrocarbons C$_{19+}$ and an increase in the content of olefins C$_{5+}$.

**Table 1.** Products-catalyst-synthesis conditions.

| Product | Catalytic Bed Organization | Zeolite | Synthesis Conditions | Ref. |
|---|---|---|---|---|
| Hydrocarbons gasoline fraction | Core-shell catalyst | HZSM-5 | 1 MPa, 280 °C, H$_2$/CO = 2, 2.24 L/(g·h) | [58] |
| | | | 1 MPa, 260 °C, H$_2$/CO = 2, W$_{cat}$/F$_{syngas}$ = 5 g·h·mol$^{-1}$' | [59] |
| | | | 2.1 MPa, 200–250 °C, H$_2$/CO = 2, 0.5 L/h | [60] |
| | | | 2 MPa, 210–260 °C, H$_2$/CO = 2, GHSV = 1000 h$^{-1}$ | [61] |
| | | | 1 MPa, 260 °C, H$_2$/CO = 2 | [62] |
| | | | 2 MPa, 250 °C, H$_2$/CO = 2 | [63] |
| | | | 1.2 MPa, 230 °C | [64] |
| | | ZSM-5 + MOR | 1 MPa, 260 °C, H$_2$/CO = 2 | [65] |
| | | HBeta | 1 MPa, 265 °C | [66] |

**Table 1.** *Cont.*

| Product | Catalytic Bed Organization | Zeolite | Synthesis Conditions | Ref. |
|---|---|---|---|---|
| Hydrocarbons gasoline fraction | Impregnation of zeolite | H-USY, H-Beta, H-Mordenite or H-ZSM-5 | 10 bar, 240 °C, $H_2/CO = 2$, GHSV = 1287 $h^{-1}$ | [69] |
| | | ZSM-5 | 1.9 MPa, 280 °C, $H_2/CO = 2$, GHSV = 750 $h^{-1}$ | [70] |
| | | HZSM-5 | 2 MPa, 235–300 °C, $H_2/CO = 2$, GHSV = 1500 $h^{-1}$ | [71] |
| | | ZSM-5 | 2 MPa, 240–260 °C, $H_2/CO = 2$, 3000 mL/$g_{cat}$/h | [72] |
| | | ZSM-5 | 2 MPa, 210 or 250 °C, $H_2/CO = 2$, 15 or 45 $cm^3$/($g_{cat}$·min) | [73] |
| | Impregnation of zeolite or mixed bed | ZSM-5, MOR or Beta | 2 MPa, 250 °C, $H_2/CO = 2$, 34 L/($g_{Co}$·h) | [75] |
| | Impregnation of zeolite | Beta | 1 MPa, 220 °C, $H_2/CO = 2$ | [76] |
| | | Cancrinite | 1 MPa, 260 °C, $H_2/CO = 2$, 6 g·h·mol$^{-1'}$ | [77] |
| | | HZSM-5 | 1.5 MPa, 220–240 °C, $H_2/CO = 2$ | [8] |
| | | HZSM-5 | 1 MPa, 240 °C | [78] |
| | | HZSM-5 | 2 MPa, 220 °C, $H_2/CO = 2$, 2.4 L/(g·h) | [79] |
| | | H-ZSM-5 | 2 MPa, 240 °C, $H_2/CO = 2$ | [80] |
| | | H-ZSM-5 | 2 MPa, 240 °C, $H_2/CO = 1$, 20 mL/min | [81] |
| Isoparaffins | Impregnation of zeolite | Y | 1 MPa, 260 °C, $H_2/CO = 2$ | [82] |
| | | Beta | 2 MPa, 225 °C, $H_2/CO = 2$, 4.8 L/(g·h) | [83] |
| | | ZSM-5 | 2 MPa, 250 °C, $H_2/CO = 2$, 20–70 L/($g_{Co}$·h) | [84] |
| | | HBeta | 2 MPa, 250 °C, $H_2/CO = 2$, 66 L/($g_{cat}$·h) | [85] |
| | Impregnation of zeolite or mixed bed | ZSM-5, MOR or BEA | 2 MPa, 250 °C, $H_2/CO = 2$, 34 L/($g_{Co}$·h) | [75] |
| Diesel fuel components | Impregnation or melt infiltration | HY or NaY | 2 MPa, 230 °C, $H_2/CO = 1$, 20 mL/min | [86,87] |
| | Impregnation of zeolite | HZSM-5 | 2 MPa, 210 °C, $H_2/CO = 2$, GHSV = 1200 $h^{-1}$ | [88,89] |
| | Composite | HZSM-5 | 2 MPa, 180–250 °C, $H_2/CO = 2$, GHSV = 1000 $h^{-1}$ | [90,91] |
| Synthetic oil | Composite | ZSM-5 or ZSM-12 | 0.5–3 MPa, 205–235 °C, $H_2/CO = 1–2$ | [94] |
| | Impregnation of composite or composite | HBeta | 2 MPa, 170–260 °C, $H_2/CO = 2$, GHSV = 1000–5000 $h^{-1}$ | [93,95, 96] |
| | Impregnation of zeolite | ZSM-5 | 1 MPa, 230 °C, | [97] |
| | | ZSM-5 | 2 MPa, 240 °C, $H_2/CO = 2$, 4 and 10 L/($g_{cat}$·h) | [98] |
| Olefins | Core-shell catalyst | NaZSM-5 | 1 MPa, 300 °C, $H_2/CO = 2$ | [104] |
| | Impregnation of zeolite | HZSM-5 | 1 MPa, 240 °C, $H_2/CO = 2$ | [78] |

Thus, it is obvious that the composition of the synthesis products depends on the number of contacts of hydrocarbons formed from CO and $H_2$ with the acid sites of zeolite (Figure 4). And this is ensured by the availability of acid zeolite sites for adsorption and further transformations of primary hydrocarbons. The most profound changes in MWD are observed when using core-shell catalysts, in which the promotion of FTS products is impossible without repeated contact with zeolite sites. In this case, the formed hydrocarbons do not contain components with a number of carbon atoms greater than $C_{12}$. The probability of contacting the freshly formed hydrocarbons with acid sites decreases with increasing the distance between zeolite acid sites and Cobalt active sites.

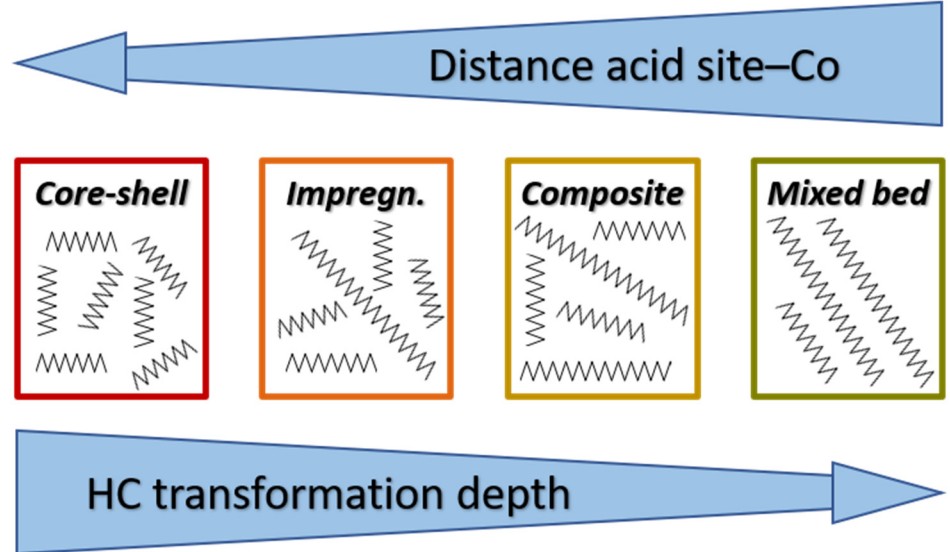

**Figure 4.** Ways to control a Co-zeolite interface: graphical representation.

The distance between cobalt and zeolite sites is also of great importance, which also determines the quantity and quality of contacts of FTS products with acid zeolite sites. Thus, in the review [10], it is noted that in the process of skeletal isomerization of n-paraffins, it is necessary to ensure a close distance between metal and acid sites to reduce the probability of side reactions with the formation of coke. However, in [105], it is emphasized that when cracking large and complex hydrocarbon molecules, the excessive proximity of metallic and zeolite sites reduces the selectivity of the process. Therefore, optimizing the distance between different types of catalytic sites is an important aim in the development of multifunctional catalysts. In work [106], a comparative study of a pellet catalyst containing skeletal cobalt and zeolite HBeta, a physical mixture of powdered catalyst components and their layered arrangement was carried out in FTS at 2 MPa in the temperature range of 170–270 °C. A comparison of the compositions of $C_{4+}$ hydrocarbons showed that the closer the metal and zeolite active sites are located to each other, the lower the average molecular weight of the products formed is, in which the content of olefins and iso-paraffins increases. The same conclusions are reached by authors of works [67,98]. It should also be noted that a number of authors note a decrease in soot formation on the surface of zeolite-containing FTS catalysts and an increase in their stable operation period, which is associated with the optimal location of active sites relative to each other.

Therefore, the structure of Co-zeolite interface can be tuned in the following ways: (1) in the case of choosing the preparation method of catalytic particles (physical mixture of FTS Co catalyst and zeolite, impregnation of zeolite, formation of core-shell fraction, where core is FTS Co catalyst and shell is zeolite, formation of FTS Co catalyst and zeolite into a single pellet); (2) changing the ratio of Co and zeolite sites; (3) varying the particle size of initial zeolite (shell thickness); (4) regulating the acidity of the initial zeolite (type, SAR, presence of different cation than $H^+$).

In the process of analyzing the literature data, we formed a hypothesis about the role of the distance between active metal and zeolite sites: it is significant to optimize it, so that the hydrocarbon molecule desorbed from the metal site retains its energy potential (excess energy) until adsorption on the acid site in order to reduce the activation energy of further transformations (Figure 5).

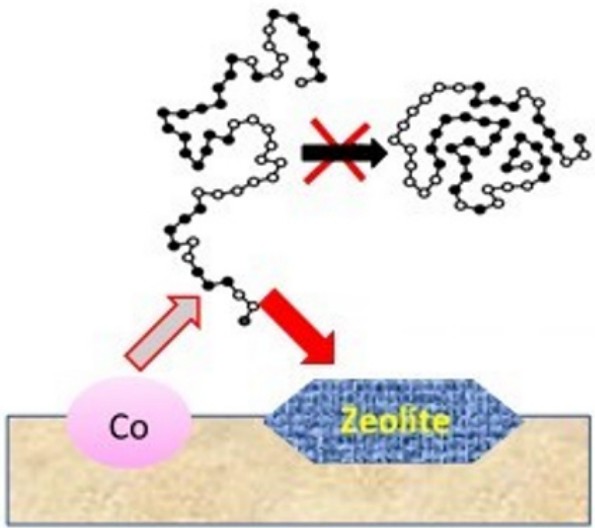

**Figure 5.** Possible ways for a hydrocarbon molecule.

The composition of the synthesis products obtained in the presence of a catalyst with an optimized distance between active metal and zeolite sites will depend on the relative rates of reactions at acid and metal sites, which in turn depend on (1) the concentration, structure and location of metal sites; (2) the strength, concentration and availability of acid zeolite sites; (3) their location relative to each other; (4) a porous system sufficiently extended to reduce diffusion limitations; and (5) the conditions of synthesis (first of all, the contact time).

In addition to the availability of acid sites and the distance between them and metal sites, some authors emphasize the role of the sizes of Co crystallites and the oxidation degree, as well as the degree of metal–zeolite interaction. In work [107], using the example of a Co/(S15 + Z5) catalyst obtained through impregnation of a mixture of SBA-15 and ZSM-5 (1:1), it was shown that on the surface of an SBA-15-based catalyst, cobalt was mainly represented by $Co_3O_4$ in the form of well-dispersed particles of 13.3 nm in size, and based on ZSM-5 was CoO, characterized by strong interaction with the support, and $Co_3O_4$ in the form of large particles with a size of 26 nm with low dispersion. According to the authors, the ratio of $Co_{2+}/Co_{3+}$ equal to 0.63 and the size of the crystallites of $Co_3O_4$ equal to 14.1 nm contributed to an increase in the selectivity of the hydrocarbons $C_{12}$–$C_{22}$ formation from 41 to 52%. In [76,78,86,87], the role of cobalt particle sizes and the interaction degree with the support in control of the selectivity of Co-zeolite FTS catalysts is also noted. The dependence on the oxidation degree of the group VIII metal in the precursor is well known and confirmed for another industrially important catalytic process, namely for the synthesis of ammonia, where the transition from $Fe_3O_4$ magnetite to FeO vustite changes the kinetics of the process [108].

## 6. Conclusions

In general, a comparative analysis of the literature data allows to formulate the main principles of catalytic particles formation active in FTS and acid-catalyzed transformations of hydrocarbons: (1) the presence of weak Bronsted acid sites to control cracking, (2) an availability of Bronsted acid sites for re-absorption hydrocarbons, and (3) a weak Co-zeolite interaction to reduce methane formation.

**Author Contributions:** Conceptualization, supervision, L.S. and V.M.; literature search and writing—original draft, L.S. and E.A.; writing—review, editing, translation, V.M. and V.S. All authors have read and agreed to the published version of the manuscript.

**Funding:** This work was supported by the Ministry of Science and Higher Education of the Russian Federation within the framework of the Assignment to Technological Institute for Superhard and Novel Carbon Materials.

**Data Availability Statement:** Not applicable.

**Acknowledgments:** The authors thank the Center for collective use of scientific equipment «Studies of Nanostructured, Carbon and Superhard Materials» of Technological Institute for Superhard and Novel Carbon Materials for support and help.

**Conflicts of Interest:** The authors declare no conflict of interest.

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
