# Peer review of "Zeolite-Containing Co Catalysts for Fischer–Tropsch Synthesis with Tailor-Made Molecular-Weight Distribution of Hydrocarbons"

_reactions, doi:10.3390/reactions4030022_

Round 1

Reviewer 1 Report

The review summarizes F-T synthesis using Co catalysts in aspects of changing the strength, concentration of the active sites and inter-site distances for tuning product distributions, and concludes the pricinples of active catalyst formation and hydrocarbons transformation. The whole paper is acceptable, minor revision is need. Some of my concerns are given below.

1. The authors discussed yje role of zeolite in the F-T process, it is better to add some contents about  the role of Co in the F-T synthesis process.

2. The approaches to adjust the acidity or the basicity of zeolite should be added, which can provide guidancelines for related pionneers conducting the same or related reactions.

3.  It is not so proper to cite references in Conclusion part. 

4. It seems that related results of cited references are missed, which make the paper hard to be fully read. It is suggested to add some more pictures during literature discussion to improve readership.

The quality of english usage is good. Minor editing of English language required.

Author Response

Point 1: The authors discussed yje role of zeolite in the F-T process, it is better to add some contents about the role of Co in the F-T synthesis process.

Response 1: Chapter 2 is devoted to the role of Co in a revised version, this chapter is named «The role of Co in the formation of Fischer–Tropsch synthesis products». In addition, Figure 2 was added to demonstrate the reactions occuting on the Co active sites under F-T process (p. 6).

Point 2: The approaches to adjust the acidity or the basicity of zeolite should be added, which can provide guidancelines for related pionneers conducting the same or related reactions.

Response 2: We thank the Reviewer for this suggestion. Appropriate information was added into the text (lines 142-147).

Point 3: It is not so proper to cite references in Conclusion part.

Response 3: We agree with this comment, thank you. The new version of Conclusion part is located in the text at lines 725-730 and does not contain references.

Point 4: It seems that related results of cited references are missed, which make the paper hard to be fully read. It is suggested to add some more pictures during literature discussion to improve readership.

Response 4: We thank the Reviewer for this comment. Table 1 (pp. 15-16.) was added into the text to demonstrate all cited results. Moreover, four new Figures 1-4 were introduced into the text (pp. 2, 6, 7, 17).

Reviewer 2 Report

I studied the review manuscript with great interests. The authors outline the advances in the structural design of Co catalysts on zeolites for FTS reactions. The structure-performance relationships of Co active sites and surface acid sites are carefully discussed. This topic is highly valuable, and therefore, this review manuscript can be considered for publication after several issues are addressed.

(1) It is necessary to clearly present the reference works with Figures.

(2) Tables are also needed to summarize the performances of different kinds of catalysts in FTS reactions with different products.

(3) Perspectives of the synthesis methods to tune the local structure of Co-zeolite interface with regulated interaction/distance should be provided in the Section of 5. Conclusions and Perspectives.

Author Response

Point 1: It is necessary to clearly present the reference works with Figures.

Response 1: We thank the Reviewer for this comment. New Figures 1-4 were introduced into the text (pp. 2, 6, 7, 17).

Point 2: Tables are also needed to summarize the performances of different kinds of catalysts in FTS reactions with different products.

Response 2: Table 1 (pp. 15-16) was added into the text to summarize the performances of different kinds of catalysts in FTS reactions with different products.

Point 3: Perspectives of the synthesis methods to tune the local structure of Co-zeolite interface with regulated interaction/distance should be provided in the Section of 5. Conclusions and Perspectives.

Response 3: The Section 5 «Possibilities for the product composition contol» and appropriate information were introduced into the text (lines 687-693).

Reviewer 3 Report

This manuscript provides an extensive review on FTS catalyzed by Co catalysts with zeolite to produce hydrocarbons with narrower MWD. A large number of studies on this topic are introduced, including different structure of the composites (Co-zeolite core-shell, Co deposited on zeolite, physical mixture, etc.) and different products (gasoline, diesel, isoparaffins, olefins, etc.). This review covers different aspects and summarized quite a lot of data, but lacks new understandings from the authors based on the existing reports. The review contains descriptions of numerous works, but only contains one schematic figure. The authors should make several tables to compare the structure of catalysts, reaction condition and the resulting MWD and type of products. The author should also add more schematic figures to illustrate the interaction between Co and zeolite with different geometrical relationship, and the mechanism of MWD control with different type of catalyst composites.

Author Response

Point:. The authors should make several tables to compare the structure of catalysts, reaction condition and the resulting MWD and type of products. The author should also add more schematic figures to illustrate the interaction between Co and zeolite with different geometrical relationship, and the mechanism of MWD control with different type of catalyst composites.

Response: Figure 3 (p. 7) was introduced to illustrate the methods of catalyst forming that provide different geometrical relationship (distence) between Co and acid sites of zeolite. The Section 5 «Possibilities for the product composition contol» was introduced into the text (lines 687-693). Also, Table 1 (pp. 15-16) was added into the text to demonstrate comparison the structure of catalysts, reaction condition and the resulting MWD and type of products. In addition, Figure 4 (p. 17) was added to demonstrate the ways to control a Cobalt-zeolite interface.

Round 2

Reviewer 1 Report

The revised manuscript can be accepted for publication.

Reviewer 2 Report

The authors have given reasonable responses to the queries from the reviewers and made appropriate revision. Therefore, the manuscript can be considered for publication. During the Proof stage, the manuscript should be carefully checked, and the grammatical errors need to be revised. For instance, Lines 194 and 196: “can occurs”; Line 304: “An impregnating FTS catalyst containing zeolite are the most common with developers of bifunctional catalysts”.

The grammatical errors need to be revised.

Reviewer 3 Report

The authors have thoroughly revised the manuscript and I have no more questions. I recommend this manuscript to be accepted by Reactions.